# Phylogenetic and biogeographical traits predict unrecognized hosts of zoonotic leishmaniasis

**Caroline K. Glidden**[1]*, **Aisling Roya Murran**[1], **Rafaella Albuquerque Silva**[2], **Adrian A. Castellanos**[3], **Barbara A. Han**[3], **Erin A. Mordecai**[1]

**1** Department of Biology, Stanford University, Stanford, California, United States of America, **2** Secretaria de Vigilância em Saùde, Ministério da Saúde do Brasil, Distrito Federal, Brasília, **3** Cary Institute of Ecosystem Studies, Millbrook, New York, United States of America

\* cglidden@stanford.edu

**Data Availability Statement:** All data and analyses can be found at https://github.com/ckglidden/leish_hosts.

## Abstract

The spatio-temporal distribution of leishmaniasis, a parasitic vector-borne zoonotic disease, is significantly impacted by land-use change and climate warming in the Americas. However, predicting and containing outbreaks is challenging as the zoonotic *Leishmania* system is highly complex: leishmaniasis (visceral, cutaneous and muco-cutaneous) in humans is caused by up to 14 different *Leishmania* species, and the parasite is transmitted by dozens of sandfly species and is known to infect almost twice as many wildlife species. Despite the already broad known host range, new hosts are discovered almost annually and *Leishmania* transmission to humans occurs in absence of a known host. As such, the full range of *Leishmania* hosts is undetermined, inhibiting the use of ecological interventions to limit pathogen spread and the ability to accurately predict the impact of global change on disease risk. Here, we employed a machine learning approach to generate trait profiles of known zoonotic *Leishmania* wildlife hosts (mammals that are naturally exposed and susceptible to infection) and used trait-profiles of known hosts to identify potentially unrecognized hosts. We found that biogeography, phylogenetic distance, and study effort best predicted *Leishmania* host status. Traits associated with global change, such as agricultural land-cover, urban land-cover, and climate, were among the top predictors of host status. Most notably, our analysis suggested that zoonotic *Leishmania* hosts are significantly undersampled, as our model predicted just as many unrecognized hosts as unknown hosts. Overall, our analysis facilitates targeted surveillance strategies and improved understanding of the impact of environmental change on local transmission cycles.

## Author summary

Leishmaniasis is a zoonotic, vector borne disease of poverty with a high burden throughout the Americas: within Latin America there are an estimated 58,500 new cases per year and 54,050 years of life lost due to disability. Although the World Health Organization has targeted leishmaniasis for elimination and control by 2030, the disease remains a

**Funding:** This project was supported by the National Science Foundation (Awards DEB-2011147 to EAM and supported CKG; EEID-1717282 awarded to BAH), the National Institutes of Health awarded to EAM (Award numbers: R35GM133439, R01AI102981, R0AI168097). EAM was additionally supported by the Stanford King Center for Global Development, the Stanford Woods Institute for the Environment, and the Stanford Center for Innovation in Global Health. ARM was funded by a small grant from the Office of the Vice Provost for Undergraduate Education at Stanford. The funders had no role in study design, data collection and analysis, decision to publish, or preparation of the manuscript.

**Competing interests:** The authors have declared that no competing interests exist.

persistent threat. Across the Americas, particularly in Central America, the southeastern United States, and perimeters of the Amazon Basin, risk of infection is increasing in geographic extent and elevation. While it is known that *Leishmania* parasites, the causative agent of leishmaniasis, are maintained in the environment via a mammalian host, the full suite of wildlife hosts has yet to be documented, which significantly hinders control efforts. Here, we use machine learning and ecological and evolutionary trait profiles of known hosts to identify unrecognized potential wildlife hosts of *Leishmania*. We identify 136 mammals in the Americas that are likely to be exposed to and infected by zoonotic *Leishmania* in the wild. The high number of unrecognized potential hosts emphasizes a need to better invest in studying the ecological epidemiology of leishmaniasis. The study provides information and tools to support targeted intervention and management of this important poverty-associated disease.

## Introduction

Leishmaniasis, a debilitating and sometimes fatal parasitic disease, is one of the 20 neglected tropical diseases targeted for control and elimination by the United Nations by 2030 [1]. However, leishmaniasis remains a persistent threat throughout the tropics and subtropics. In the Americas, leishmaniasis cases occur from the southern United States to Argentina, including the Caribbean, with an estimated 58,500 new cases each year [2]. Further, incidence is rapidly shifting in geographic extent with land-use and climate change [3–7]. For instance, hotspots of leishmaniasis have significantly expanded throughout Costa Rica in the last two decades, in concert with agricultural intensification, deforestation, and reforestation [3].

Treatment for leishmaniasis has many side effects and are extremely expensive [2,8], thus control and elimination hinges on limiting transmission. *Leishmania* parasites are transmitted by sandflies (Order: Diptera; Family: Psychodidae) and maintained in the environment via a multitude of sylvatic and domestic mammalian hosts. As such, controlling transmission requires ecological interventions targeting either component of the transmission cycles. For instance, clearing vector habitat proximate to households, supporting populations of natural enemies that regulate populations of vectors or small mammals (e.g., fish and mosquito larvae), and/or modifying distribution of wildlife food sources are sustainable and effective interventions in preventing transmission of flaviviruses, henipaviruses, *Plasmodium* spp. (malaria), and *Brucella abortus* (brucellosis) [9,10]. Similar interventions could be employed against *Leishmania*, however, successful application requires species identification of vectors, hosts, and their corresponding interactions with their biotic and abiotic environment.

Leishmaniasis was first diagnosed by Western medicine in the Americas in 1909, at the time dogs and foxes were believed to be the zoonotic hosts [11]. Continued research throughout the century hypothesized that rodents, and in some cases opossums and sloths, were the primary zoonotic hosts of these pathogens [11]. However, especially with the advancement of molecular diagnostic techniques, the number and taxonomic diversity of hosts has since expeditiously increased. In the Americas, there are now 137 species of wildlife that are recognized as potential hosts (i.e., animals that are naturally exposed and become infected once exposed), of which around 60 may act as competent hosts (i.e., animals can become infectious once infected and may maintain the pathogen in the environment as reservoir hosts) [12,13]. Wild hosts are now known to include eight orders of mammals including Chiropetra (bats), Carnivora, Cingulata (armadillos), Didelphimorphia (opossums), Lagomorpha (rabbits), Pilosa (sloths and anteaters), Primates, and Rodentia. However, even with progress in host discovery,

the full suite of *Leishmania* hosts is not yet described: *Leishmania* spillover occurs in absence of known hosts and new hosts are discovered almost annually [14–17].

Limited knowledge on the transmission cycle makes it difficult to understand the mechanisms driving *Leishmania* distribution and spillover. As previously stated, unknown transmission cycles impede effective ecological interventions to reduce spillover risk to people. Additionally, global change can alter host dynamics to amplify spillover risk. For example, land-use change may increase host densities and/or push hosts into closer proximity to anthropophilic vectors and humans [9]. Climate warming may also shrink or expand the range of hosts, further changing the rate of contact among hosts, vectors, and humans [18]. As such, the incomplete classification of possible *Leishmania* hosts could lead to unexpected emergence and hotspots of human leishmaniasis.

Study designs optimal for detecting *Leishmania* hosts, such as systematic sampling of large populations of a diversity of species through space and time, are limited by logistics and cost of sampling. *Leishmania* host discovery could benefit from strong *a priori* hypotheses about which mammals to target for sampling effort; taxa specific sampling could help to efficiently use time and monetary resources, ultimately expediting the full classification of the *Leishmania* transmission cycle. Through the last decade, machine learning approaches have been used to generate predictions of hosts of a multitude of different pathogens, including betacoronaviruses [19], flaviviruses [20], Rabies virus [21], and *Borrelia burgdoferi* (causative agent of Lyme disease) [22]. These methods have facilitated more targeted laboratory and field work of novel pathogens and have great potential to improve our understanding of diseases of poverty with broad host and vector ranges.

Host status for any parasite or pathogen is driven by two processes: exposure and physiological susceptibility [23,24]. Exposure depends on the environmental conditions needed to support pathogen transmission and, in the case of vector-borne disease, vector reproduction, contact with competent hosts, and vector survival [25]. Following exposure, physiological susceptibility then depends on pathogen interaction with host cells to gain entry and, in some cases, avoid immune attack and replicate [25]. Host traits related to exposure and/or susceptibility likely interact in non-linear and higher-order combinations to delineate zoonotic *Leishmania* hosts from non-hosts. In brief, the effect of a trait on host status may vary in magnitude or direction depending on trait value and traits may become more or less important under different conditions. Hypothetically, for example, the average temperature of a species range may have a positive effect on host status between 25–30°C, but a negative effect outside of this range. Depending on biting activity of vector species, animals that forage on the ground may be more likely to be hosts if they are also nocturnal, whereas animals that are arboreal may be more likely to be hosts if they are crepuscular but less likely to be hosts if they are nocturnal. Thus, when viewing the zoonotic *Leishmania* system in one or two dimensions (e.g., only considering phylogeny or habitat use), it is difficult to determine unifying traits useful for identifying hosts. To date, it has been challenging to combine these processes across scales from physiological to evolutionary to ecological to biogeographic to predict their overall impact on infection potential. Machine learning offers a set of new tools that can be used to incorporate and dissect this complexity by increasing the dimensionality in which the *Leishmania* system can be studied, allowing for multiple predictors that may have nonlinear and interactive effects on host status. The flexibility of these models allows for pinpointing combinations of mammalian traits unique to the confluence of *Leishmania* exposure and susceptibility by learning patterns of known hosts. With this information, we can identify likely host species as targets for *Leishmania* surveillance efforts and the key traits that distinguish host species from non-hosts.

Here we use tree-based machine learning (extreme gradient boosted regression, 'Xgboost') and mammalian traits, including but not limited to biogeographical, phylogenetic, and life-

history features, to: (i) describe trait profiles of hosts for zoonotic *Leishmania (Viannia)* and *Leishmania (Leishmania)* parasites; and (ii) use the trait profiles to predict unrecognized wild hosts of zoonotic *Leishmania* within the subgenera *Leishmania (Viannia)* and *Leishmania (Leishmania)*.

We aim for this analysis to identify animal species that are likely to be exposed and infected in the wild; due to extreme data sparsity, we do not employ our methods to identify competent hosts (i.e., animals that become infectious once infected and may act as reservoirs given the density and distribution of their populations). Our findings can be used to identify hosts that can then be surveyed at larger spatio-temporal scales, investigated for competence, or used as sentinel species. As such, while our modeling approach does not necessarily identify competent hosts that can transmit zoonotic *Leishmania* on to sandflies and eventually humans, it does identify animals likely to be exposed to and susceptible to zoonotic *Leishmania* infection, facilitating the first step of determining epidemiological importance and fine-tuning surveillance efforts.

## Methods

### Data collection

Leishmaniasis is caused by protists within the *Leishmania* genus. There are 14 known species of zoonotic *Leishmania* (i.e., they spill over from animals to humans) in the Americas. These *Leishmania* species are divided between two subgenera: *Leishmania* (*Viannia*) and *Leishmania (Leishmania)*. In the Americas, *Leishmania (Viannia)* consists of ten species, nine of which are zoonotic; *Leishmania (Leishmania)* consists of seven species, five of which are zoonotic [26]. *Leishmania (Viannia)* spp. cause cutaneous and mucocutaneous leishmaniasis, which manifests as skin lesions that are susceptible to painful secondary bacterial infections, and/or destruction of the mucus membranes of the nose, mouth, and throat [2]. In addition to cutaneous and mucocutaneous leishmaniasis, species within the *Leishmania (Leishmania)* subgenera (*L. (L.) infantum* and, less frequently, *L. (L.) mexicana*) cause fatal visceral leishmaniasis. Visceral leishmaniasis affects internal organs, typically causing enlargement of the spleen and liver [2]. The following analyses were conducted for each subgenus (*L. (Viannia)* and *L. (Leishmania)*) so to better described the ecological associations of each taxonomic group of parasites and their corresponding disease manifestations.

### *Zoonotic* Leishmania *host status*

We gathered *Leishmania* host status from all wild, endemic, and invasive terrestrial mammals with ranges in Mexico, Central America, and South America using recent reviews [12,13] in addition to a Web of Science query on February 12, 2021 and February 22, 2021 (see S1 Text for the specific search strings). Host status was defined as a binary trait: an animal has been naturally infected by one of the 14 *Leishmania* species known to cause leishmaniasis in humans (1) or there is no record of infection by a *Leishmania* species known to cause leishmaniasis in humans (0). The latter category ("non-positives") includes animals that have been tested for *Leishmania* and have not been found to be infected as well as those that have not been tested for *Leishmania* infection. Ideally, the analysis would include true-positives (1) and true negatives (0). However, due to the intensive, systematic sampling required to declare an animal as a true-negative (longitudinal sampling across large geographic scales), there is not enough data to conduct the analysis with true-negatives. Our analysis (using true-positives and non-positives) still enables us to identify species that have not yet been tested but have a high probability of being a host. Infection was determined via detection of *Leishmania* species specific antigens or genetic material. Species identification was performed either using animal

tissue samples directly or using live parasites cultured from animal tissue. To ensure that all relevant mammals were included in our study, a list of species names of endemic Latin American mammals was retrieved from the International Union for Conservation of Nature (IUCN) Red List Inventory [27] and a list of invasive mammals was retrieved from the Global Invasive Species Database [28]. We cross-referenced our host status data table with the Global Infectious Disease and Epidemiology Network [29]. We then limited our analysis to wild mammals that had at least a 10% range overlap with reported human cases of leishmaniasis for inclusion in our analysis. In total, 86.64% (1460/1685) of species were retained for the *L. Viannia* analysis and 87.24% (1470/1685) of species were retained for the *L. Leishmania* analysis. Zoonotic *Leishmania* ranges were constructed by outlining a concave polygon around cases of disease occurrence in humans [30] (Fig 1A and 1C). We conducted all data filtering and downstream analyses separately for each subgenus.

## Traits associated with Leishmania exposure and susceptibility

Our analysis leverages patterns of traits of known hosts to predict identities of unrecognized hosts. We used host traits related to *Leishmania* exposure (via sandfly bites) and susceptibility, including life-history, biogeographical, and phylogenetic traits (S1 Table).

For life-history and some habitat use traits, we collected traits from panTHERIA [31], mammalDIET [32], and EltonTraits [33]. We additionally used IUCN range maps and satellite imagery to extract data associated with species biogeography, including variables related to climate [34,35], land-cover type [36], and global human modification index, an aggregate measure of anthropogenic pressure on a landscape [37], within each species range (see S1 Table for full description of traits and geospatial datasets). Ranges of endemic species were retrieved from the IUCN database and ranges of invasive species in Mexico, Central America, and South America were constructed by building a concave polygon around species occurrence points from [38]. *Leishmania* and invasive mammal species ranges were built using the R (v 4.0.2) packages *speciesgeocodeR* [39] and *concaveman* [40]. We additionally included IUCN-designated main habitat and habitat breadth (IUCN) as traits. To quantify phylogenetic traits, we downloaded pairwise divergence time (phylogenetic distance) from TimeTree.org [41] and reduced the dimensionality of this matrix by mapping species in ordination space using principal coordinate analysis in base R. TimeTree is a knowledge-base that has collected and synthesized species divergence times from > 3000 peer-reviewed studies. We included whether an animal is invasive or endemic in the given *Leishmania* range and zoonotic host status using GISD and GIDEON. Finally, we accounted for sampling effort by downloading the number of citations found on PubMed per species using the R package *easyPubmed* [42]. We choose to use PubMed as this indicates the biomedical study effort, as opposed to a more general measure of study effort that would be estimated using the number of citations on Web of Science. Biomedical study effort better represents the number of times a species was studied in the context of testing for pathogen assemblages.

## Data analysis

We applied extreme gradient boosted regression (XGboost) in the R package *xgboost* [43] to use mammal trait data to predict the probability that a species is a *Leishmania (Leishmania)* or *Leishmania (Viannia)* host. Extreme gradient boosted regression is a machine learning algorithm that creates an ensemble of weak decision trees to form a stronger prediction model by iteratively learning from weak classifiers and combining them into a strong classifier (i.e., boosting) [44]. Gradient boosted regression is flexible in that it allows for non-linearity, both among features (i.e., interactions) and between features and predictions, collinearity between

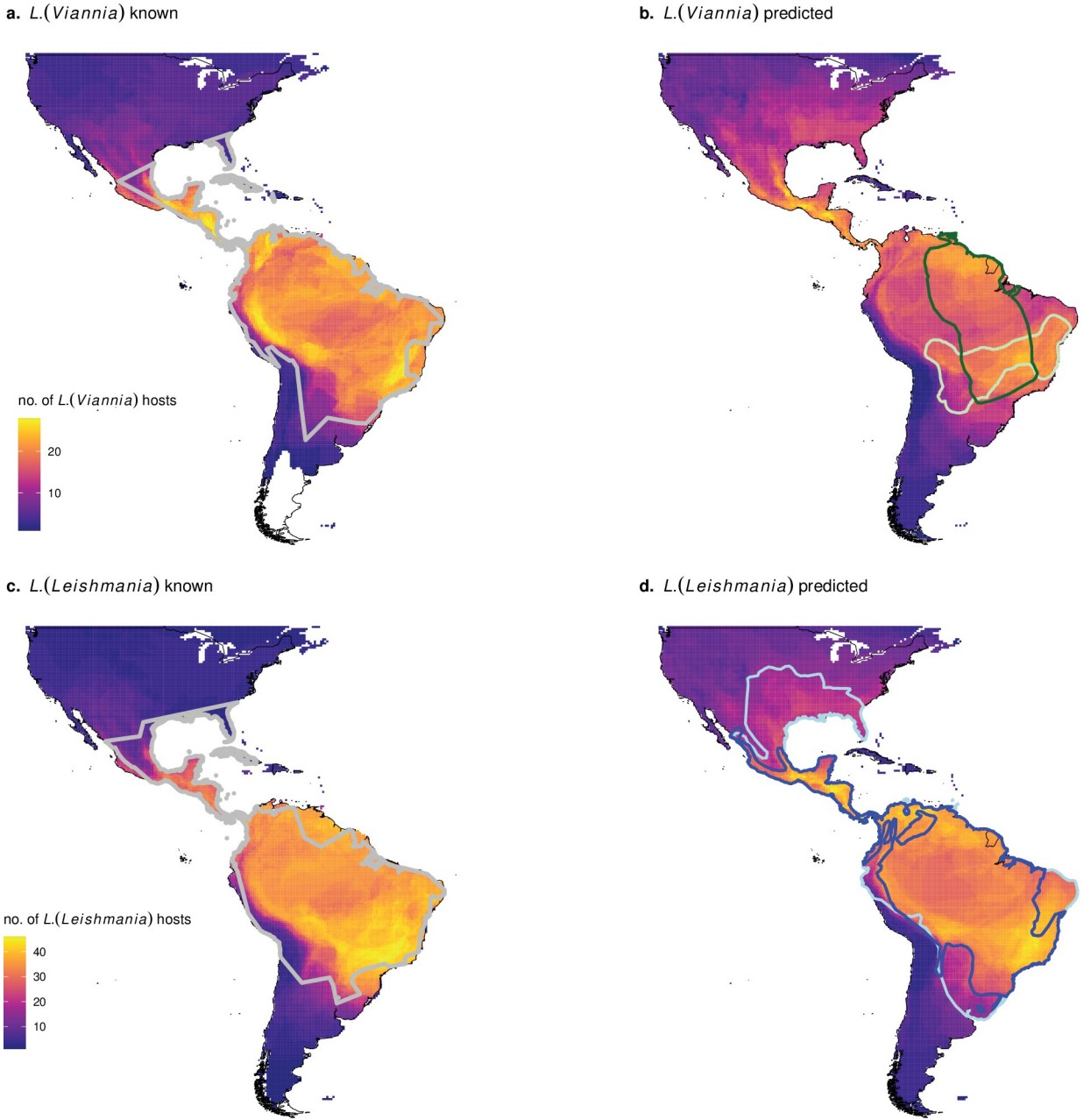

**Fig 1. The ranges of zoonotic *Leishmania* and their known and predicted hosts extend throughout the Americas.** The number of overlapping ranges of known *Leishmania* hosts are depicted in (a & c), with the range of human cases of *L. (Viannia)* and *L. (Leishmania)* outlined in grey. The number of overlapping ranges of newly predicted hosts are outlined in (b & d). *Hylaeamys megacephalus* and *Calomys callosus* are unrecognized hosts with the highest and second highest predictions for *L. (Viannia)* according to Shapley values. Their ranges are outlined in dark green and light green, respectively, in (b). *Dasypus novemcinctus* and *Leopardus wiedii* are hosts with the highest and second highest Shapley values for *L. (Leishmania)*. Their ranges are outlined in dark blue and light blue, respectively, in (d). An exhaustive list of newly predicted hosts is listed in S2 Table. Base maps were created in R using open source shapefiles from Natural Earth (admin-0 countries: https://www.naturalearthdata.com/downloads/10m-cultural-vectors/; https://www.naturalearthdata.com/about/terms-of-use/) [51]; species ranges were created in R using open source shape files from IUCN (https://www.iucnredlist.org/resources/spatial-data-download; https://www.iucnredlist.org/terms/terms-of-use) [27].

features, and non-random patterns of missing data. Further, XGboost handles extremely unbalanced data well by weighting positive labels to increase class separability and allowing for a regularization parameter to prevent overfitting to a few positive labels–an advantage when

analyzing our data set with relatively few known hosts compared to the number of mammal species examined. The ability to handle non-random patterns of missing data and extremely unbalanced data makes the XGBoost algorithm best suited for our study as compared to other machine learning methods, such as random forest and support machine vector, which overfit unbalanced data (random forest) and/or cannot handle missing data without use of imputation (random forest, support machine vector). Prior to analysis we removed traits with $> 0.7$ correlation to increase interpretability of variable importance. For the *L. Viannia* analysis 32/66 traits were retained in the final analysis; for the *L. Leishmania* analysis 33/66 traits were retained. To maximize the number of traits retained in the model, out of a pair of correlated traits, the one with the highest number of correlations to other traits and/or the lowest data coverage was dropped. We used a box-cox transformation of extremely skewed variables to reduce the influence of extreme values on model performance. Categorical traits were one-hot-encoded so that each category was represented as a binary trait. Finally, traits with $< 10\%$ coverage (i.e., less than 10% of the species had data on the trait) were removed and not used in the analysis (for training the model and prediction).

## Model performance and hyperparameter tuning

The accuracy of our model was evaluated using nested cross-validation. Nested cross-validation, as opposed to K-fold cross-validation, produces the least biased evaluation of model performance when using a small sample size [45]. Three-fold cross validation was used to generate model predictions and evaluate overall model performance, while 5-fold cross validation was performed to tune hyperparameters within each 3-fold cross validation step. In other words, the full dataset was split into three outer folds. Hyperparameters were tuned using Bayesian optimization, optimizing over a parameter space aimed to reduce overfitting (low training rate, low ratio of samples used in trees, low ratio of features used in trees, low maximum tree depth, and high regularization) using a 5-fold cross-validation procedure within the training data. Using the hyperparameters that yielded the best performance (minimum logloss), a final model was then trained using the full data within these two folds and model predictions were calculated on the third hold-out fold. This process was repeated until out-of-sample predictions were made for each fold. As model output may be dependent on the distribution of the data across each fold, nested cross validation was repeated 25 times using 25 unique splits of the data. Our dataset was extremely unbalanced (i.e., many more 0s than 1s), which makes our model vulnerable to overfitting. To determine if our model was simply fitting spurious correlations in the data, we conducted a target shuffling analysis [46,47]. We repeated the nested cross-validation analysis but randomized the response variables (*Leishmania* host status) for each iteration. We then calculated the mean AUC from the target shuffling experiment. AUC (area under the receiver operator characteristic curve) is a classification metric that measures the probability that model output for a randomly chosen positive label (known host) will be higher than a randomly chosen negative label (animal with no record of zoonotic *Leishmania* infection). AUC ranges from 0 to 1, with an AUC of 1 indicating that the model perfectly classifies all samples, while an $AUC < 0.5$ indicates the model performs no better than a coin flip. We found that our model performed minutely better than a coin flip in the target shuffling experiment (average *L. (Leishmania)* target-shuffled AUC = 0.53; average *L. (Viannia)* target-shuffled AUC = 0.54), thus we adjusted our final AUC values by $AUC_{\text{final model}} - (AUC_{\text{target shuffled model}} - 0.5)$.

In our final model, citation count was among the top 10 most important variables. To ensure that we were not simply predicting well-studied animals, we repeated the analysis above while replacing the response variable with study effort (the number of citations on PubMed). To reduce computational time, we ran ten iterations instead of 25. We concluded

that our model was not simply predicting well-studied species by evaluating how well the traits in our model predict study effort, measured as pseudo-$R^2$. Additionally, we checked that trait responses (i.e., partial dependence plots) that predicted publication number were not overly similar to the functional form of the trait responses that predicted host status.

## Leishmania *host trait profiles*

To assess *Leishmania* host trait profiles, we first identified important features using SHAP (Shapley Additive Explanations) calculated via the R package *SHAPforxgboost* [48]. Shapley scores represent the average marginal contribution of a feature to a prediction across all combinations of features; the contribution of the feature is interpreted as the change in prediction associated with that feature in relation to the average model prediction [44]. SHAP calculates local feature contribution by evaluating the contribution of each feature to the prediction for each observation in a dataset. Here, local contribution refers to the contribution of each feature to the prediction for each individual species (local corresponds to the level of a single sample, which, for our analysis, is a species). Global feature contribution can be obtained by aggregating the Shapley scores for each feature across all observations; global feature contribution is the average contribution of each feature across all species (global corresponds to the level of the entire dataset). To generate a measure of uncertainty in feature contribution, we trained the model with 70% of the data and calculated Shapley scores for the remaining 30% of observations. We repeated the above procedure 100 times, using different subsets of data with each iteration. We then described trait profiles using SHAP partial dependence plots, which maps the relationship between the contribution of the feature to model predictions for each value of the feature in the dataset.

## Leishmania host predictions

Following [21], we identified animals as unrecognized zoonotic *Leishmania* spp. hosts using the Shapley value classification criterion. The sum of the Shapley scores across features for each species sums to the prediction for each species relativized by the average model prediction across species. A summed Shapley value of 0 represents the average prediction in the dataset [49,50]. To account for model uncertainty, we used a bootstrapping procedure to calculate Shapley scores for models fit with 70% of the data 100x. We classified an animal as a newly predicted host if > 95% of Shapley scores were > 0; in other words, we classified an animal as a newly predicted host if predicted probability of being a host was greater than the average probability for that model iteration for > 95% or model iterations [21].

The full record of species infection status and references, trait data-base and R code can be found at https://github.com/ckglidden/leish_hosts.

## Results

### Leishmania (Viannia)

Through our literature search, we found 96 known *L. (Viannia)* hosts out of 1460 mammals that occur within the geographic range of *L. (Viannia)* parasites (Figs 1A and 2A). After accounting for target shuffling, our model performed moderately well with an average out of sample AUC of 0.81 (95% CI: 0.80–0.82) and in sample AUC of 0.94.

### L. (Viannia) *trait profiles*

After removing highly correlated traits, 30 features were used to predict *L. (Viannia)* hosts. A mix of biogeography, life-history, phylogeny, and study effort covariates were among the top

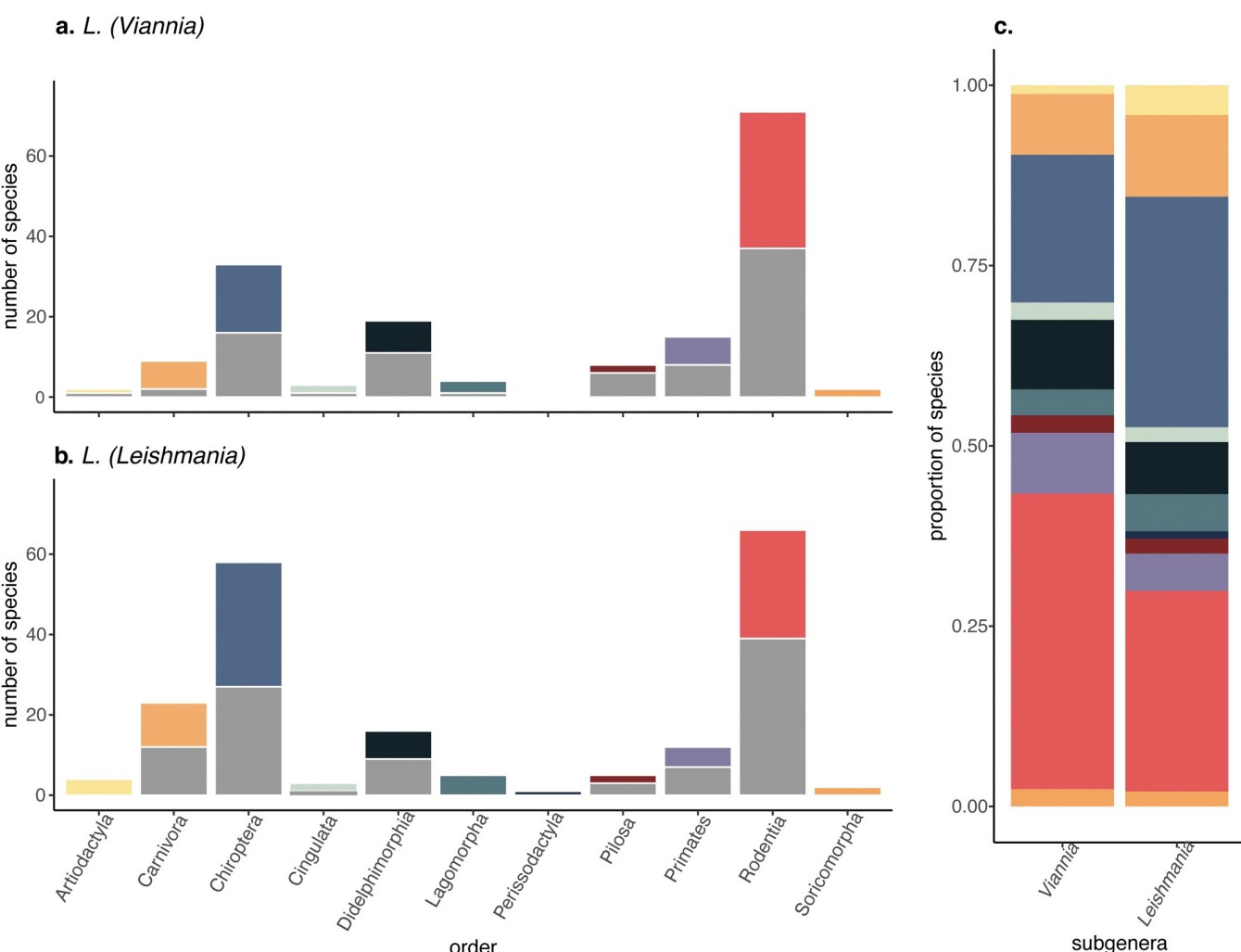

**Fig 2. The number of known and predicted hosts per order for *L. (Viannia)* (a) and *L. (Leishmania)* (b).** Grey bars represent the number of known hosts within each order, colored bars represent the number of newly predicted hosts per order. Animals were classified as newly predicted if > 95% of mean SHAP values were greater than 0 (a-b). Bars in (c) represent the proportion of newly classified hosts within each order; colors match (a-b).

important model features when estimating global feature contribution (Fig 3). Phylogenetic distances, reduced into six axes of a PCoA ordination, were, on average, the most important trait for *Leishmania (Viannia)* hosts. Specifically, animals were more likely to be hosts if they were rodents and opossums, and less likely to be hosts if they were non-human primates, carnivores, and even toed ungulates (S1 and S2 Figs). Similar to other zoonotic pathogen systems, *L. (Viannia)* hosts live at high population densities and demonstrate signatures of fast-paced life-histories, indicated by short gestation periods (Fig 4A). Further, *L. (Viannia)* host ranges are large, have high proportions of crop and urban land cover, and, on average, peak at ~28°C during the warmest quarter of the year (Fig 4A). Geographic coordinates of the species' native ranges were also an important predictors: animals were more likely to be hosts if the minimum longitude of their native range was around western Central America. Notably, although study effort (number of PubMed citations, normalized with a Box-Cox transformation) was among the most important traits, our supplementary analyses indicated that the host models were not simply predicting well-studied hosts. Specifically, the supplementary model predicting study

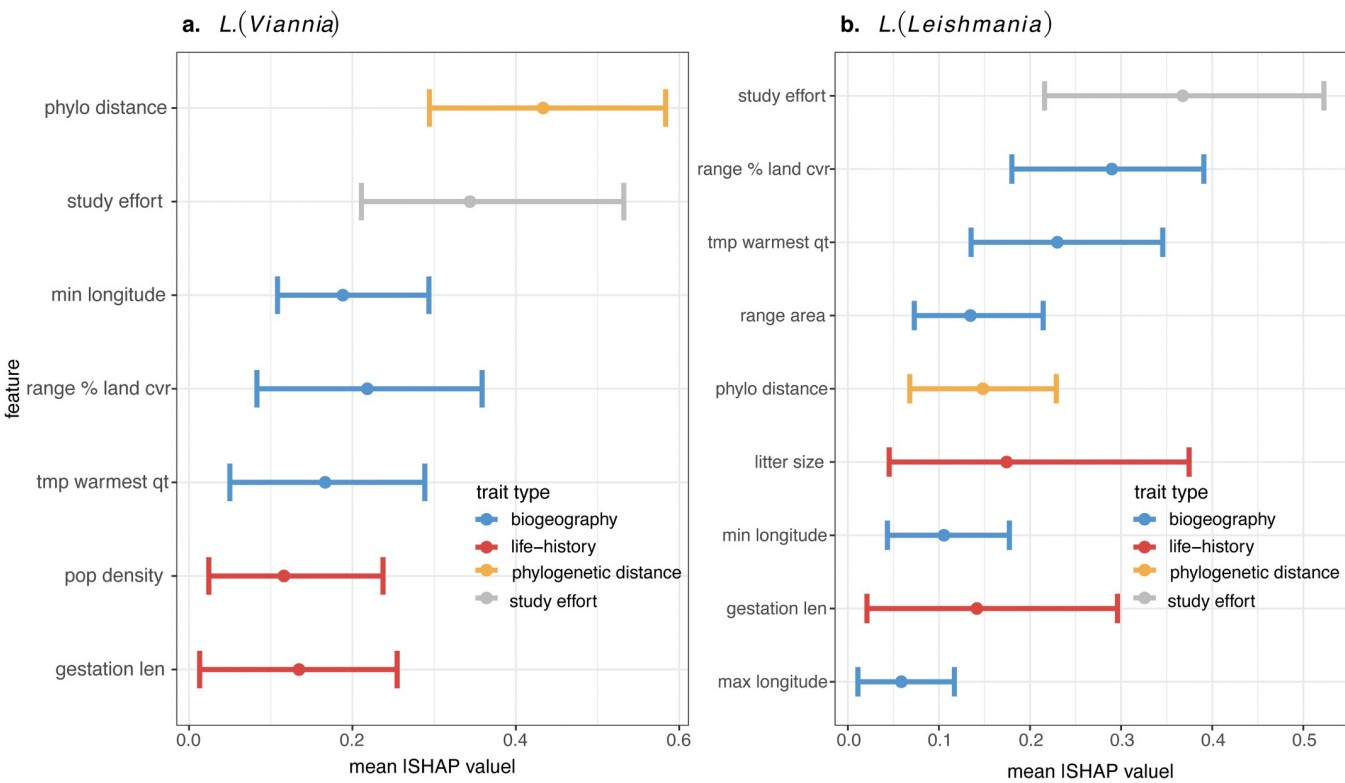

**Fig 3. Biogeography, life-history, and phylogenetic traits all significantly contributed to model predictions for host status.** Biogeographical traits (minimum longitude of the range, maximum longitude of the range, % cover of land-use/land cover in the species range, average temperature in the warmest quarter in the species range, range area) are colored in blue, life-history traits (population density, gestation length, litter size) are colored in red, phylogenetic distance (location along PCoA ordination axes) is colored in orange, and study effort is colored in grey. Points are the absolute value of the mean Shapley importance (mean |SHAP value|) for the trait across all mammals (i.e., global feature contribution), bars represent the absolute values of the 0.05–0.95 percentiles. Only features with 0.05 percentiles > 0 are shown.

effort (number of PubMed citations) performed poorly (average out-of-sample $R^2$ = 0.19). Further, traits that predict citation count, and the response curves of those traits, greatly differed from traits that predict *L. (Viannia)* host status (S3–S4 Figs). As such, while *L. (Viannia)* hosts are likely understudied, we are not simply reporting the trait profiles of well-studied mammals.

## L. (Viannia) *hosts*

Eighty-three animals were identified as likely hosts (Fig 2 and S2 Table). Out of these newly predicted hosts, the majority were rodents, bats, and opossums. Further, 14 of these hosts are known hosts of *L. (Leishmania)* (S2 Table). The newly predicted hosts with the highest summed Shapley scores were the large headed rice rat (*Hylaeamys megacephalus*) and the large vesper mouse, *Calomys callosus* (Fig 1). In contrast to known hosts, the range of newly predicted hosts extends to the southern tip of South America and there is a high density of species ranges that overlap in the southern United States (Fig 1A and 1B).

## Leishmania (Leishmania)

Through our literature search, we found 110 known hosts out of 1470 mammals that have overlapping ranges with the zoonotic *L. (Leishmania)* range (Fig 1C). Of these, 40 animals

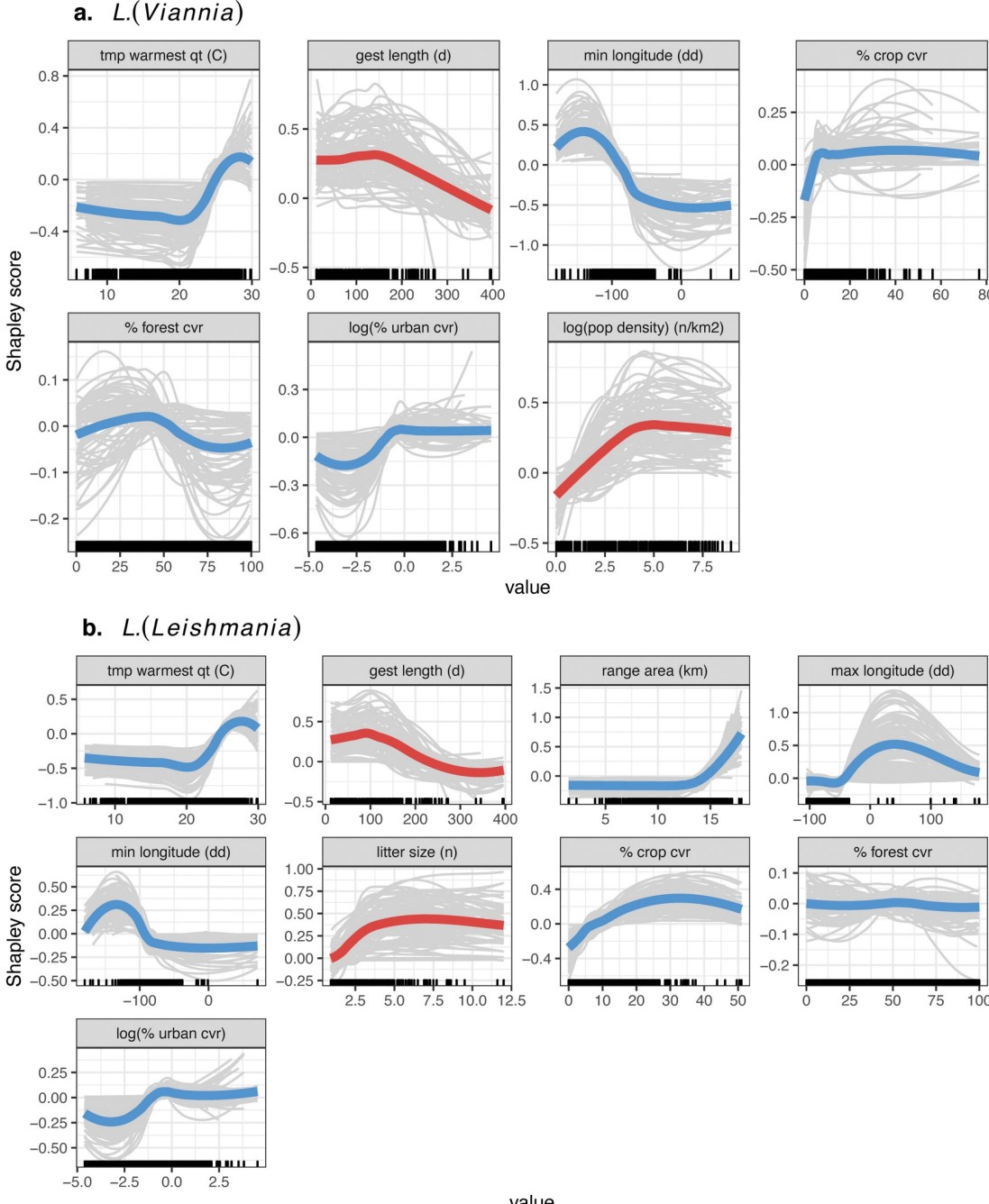

**Fig 4. Hosts have fast-paced life-histories and live in proximity to humans.** Shapley partial dependence plots showing the effect of each feature on *L. (Viannia)* (a) and *L. (Leishmania)* (b) host status after accounting for the average effect of the other features in the model. Colored lines represent the average effect across model iterations, while grey lines show each individual model iteration (model fit with 80% of data) (blue = biogeographical traits; red = life-history). Features with global mean feature contribution scores > 0 for > 95% of model iterations are shown. Rug plots on the x-axis indicate the distribution of the data.

were hosts for both *Leishmania* subgenera. After accounting for target shuffling, our *L. (Leishmania)* hosts model performed moderately well with an average out of sample AUC of 0.84 (95% CI: 0.83–0.85) and in sample AUC of 0.94.

## L. (Leishmania) *trait profiles*

After removing highly correlated traits, 32 features were used to predict *L. (Leishmania)* hosts. On average, study effort was the most important trait (i.e., had the highest global feature contribution), followed by the land cover/land-use composition of the species range, average temperature in the warmest quarter of the species range, range area, phylogenetic distance, litter size, gestation length, and longitudinal extent of the species range (Fig 3). Similarly, to *L. (Viannia)* hosts, *L. (Leishmania)* hosts show signatures of fast-paced life histories, indicated by short gestation lengths and large litter sizes, and have ranges with high degrees of crop and urban cover (Fig 3B). Further, the range of *L. (Leishmania)* hosts are similar to *L. (Viannia)* hosts in that the mean temperature in the warmest quarter peaks around 28˚C (Fig 4B). On average, geographic coordinates of the species' native range were also an important predictor: Animals were more likely to be hosts if the minimum longitude of the range was around western Central America, while the maximum longitude of the range was western Europe, which indicates a role of invasive species in the *L. (Leishmania)* transmission cycle. While study effort was among the most important traits, host traits explained little variation in number of PubMed citations (average out-of-sample $R^2$ = 0.17) and traits that predict citation count, and the response curves of those traits, greatly differed from traits that predict *L. (Leishmania)* host status suggesting that traits important for discriminating host status were not just those that predicted study effort (S5–S6 Figs).

## L. (Leishmania) *hosts*

Using the Shapley classification criterion, 98 animal species that are currently unrecognized hosts were labeled as newly predicted hosts (Fig 2 and S2 Table). Out of these newly predicted hosts, the majority were rodents, bats, and carnivores. Nine newly predicted hosts are known hosts of *L. (Viannia)*. Forty-five newly predicted hosts were newly predicted hosts for both the *L. (Leishmania)* and *L. (Viannia)* subgeneras (S2 Table). The newly predicted hosts for the *L. (Leishmania)* with the top summed Shapley scores included the nine-banded armadillo (*Dasypus novemcinctus*) and margay (*Leopardus wiedii*) (Fig 1). The nine-banded armadillo is a known host of *L. (Viannia)* parasites, with a range that extends from northern Argentina to the central United States (Fig 1D).

## Discussion

Our literature search identified 137 wild mammals in the Americas as known zoonotic *Leishmania* hosts and our model predicted there to be an additional 136 likely hosts (animals that are likely to be exposed and infected in the wild, but not necessarily infectious upon infection). The majority of the predicted hosts for both subgenera fell within the Rodentia order (*L. (Leishmania)* N = 27; *L. (Viannia)* N: = 34), with a similar number of predicted *Leishmania (Leishmania)* hosts within the Chiroptera order (N = 32). While their contribution to model predictions varied between subgenera, key traits included phylogenetic distance, human associated land-use, climate, population density, and study effort.

For both *Leishmania* subgenera, biogeographical features encompassing descriptors of species ranges were among the top five contributors to host predictive accuracy. Notably, zoonotic *L. (Leishmania)* hosts are more likely to be animals with larger ranges than non-hosts. Geographic range area was also reported to be a shared predictive feature among zoonotic hosts and reservoirs in other systems [21,53]. Further, ranges of *L. (Leishmania)* and *L. (Viannia)* hosts are made up of higher proportions of crop cover and urban cover than ranges of mammals that are not *Leishmania* hosts. These results are in accordance with previous findings that, in general, zoonotic host richness and relative abundance increases with anthropogenic

pressure [54]. Further, our results align with recent literature showing that risk of leishmaniasis is associated with agricultural intensification and urbanization [4,55]. *L. (Viannia)* hosts are marginally more likely to have ranges with moderate forest cover than non-hosts whereas there is little effect of forest cover on *L. (Leishmania)* host status. Vectors of cutaneous leishmaniasis, which is caused by all species of zoonotic *Leishmania* except *L. infantum*, are more likely to live in forests with high integrity and low human modification [56], while hosts are more common in disturbed landscapes [57]. Our analysis adds support to the latter finding, highlighting that transmission of parasites associated with cutaneous leishmaniasis likely occurs at forest-human settlement interfaces. The incomplete overlap between vector ranges and host ranges may explain the highly localized dynamics of cutaneous leishmaniasis spillover (i.e., transmission from wildlife to humans via the sandfly vector) [6,58]. Spillover occurs when reservoir hosts, vectors, and humans overlap in space and time so that forward transmission occurs from animal to sandfly to human. If vector and host distributions only overlap at habitat interfaces, area of reservoir host, sandfly, and human co-occurrence may be low and variable through time. Finally, average temperature of the warmest quarter was an important predictor of host status, with the feature contribution to model predictions peaking around 28˚C. In the case of zoonotic leishmaniasis, temperature is likely a predictor of *Leishmania* exposure as this trait profile reflects temperatures associated with vector survival and parasite development [59,60].

The profiles of important life history traits for *Leishmania* hosts resembled those observed in other comparative analyses of zoonotic diseases. Specifically, for both *Leishmania* subgenera, the probability of host status peaked at low to median lengths of gestation, a pattern similar to that of rodent reservoirs of zoonotic pathogens and competent hosts of zoonotic tick-borne pathogens [53,61]. Further, the probability of zoonotic *L. (Leishmania)* host status increased with litter size, a trend that parallels that of both rodents reservoirs of zoonotic pathogens and bat and carnivore reservoirs of rabies virus [21,53]. Altogether, our findings suggest that, like other zoonotic reservoirs and hosts, zoonotic *Leishmania* hosts may have fast-paced life-histories. Animals with fast-paced life-histories are believed to maximize fitness by expeditiously investing in short-term reproductive effort—such as through younger ages at sexual maturity, short gestation lengths, greater litter sizes—at the cost of long-term survival [62]. As a result, animals with fast-paced life histories are thought to divert energetic resources away from effective adaptive immunity [63], or utilize responses conducive to tolerance but not resistance [64], allowing them to allocate maximal resources to short term reproductive effort. As such, animals with fast-paced life-histories may be more susceptible to infections.

However, *L. (Viannia)* hosts only had one trait (gestation length) clearly associated with a fast-paced life history, suggesting weaker evidence for the "fast-paced" host hypothesis for this subgenus. Phylogenetic distance was the most important trait for *L. (Viannia)* hosts. We hypothesize that phylogenetic relationships may better account for host physiological defenses against *Leishmania* than life-history traits. If this is the case, inclusion of specific genetic markers associated with *Leishmania*-host cell interactions may improve model predictions.

Notably, study effort significantly contributed to host predictions for both subgenera. While our follow up analyses do not indicate that model results are biased by uneven study effort, this finding may suggest that zoonotic *Leishmania* hosts are generally under sampled. Though leishmaniasis is a neglected tropical disease primarily impacting resource-poor communities, state-of-the-art zoonotic *Leishmania* diagnostic and surveillance techniques are available [16,57,65–69]. As such, under sampling *Leishmania* hosts primarily stems from both the lack of funding to employ these tools during large scale surveillance efforts and the large number of animal species that are potentially hosts. The main leishmaniasis research funding mechanism within the USA, the National Institute of Health (NIH), primarily focuses on the

molecular biology of leishmaniasis work and does not prioritize research elucidating the ecology of the sandfly-wildlife transmission cycle. In fact, a simple search for "leishmania" on the NIH and National Science Foundation grants database yields a combined total of three awards aiming to elucidate transmission cycles of *Leishmania* in the Americas (searched October 10, 2022). Researchers across Latin America are faced with similar, if not more severe, funding constraints. For example, in Brazil, a major hotspot of leishmaniasis, the federal administration from 2019–2022 cut 90% of federal science funding [70]. While our model results facilitate more targeted field sampling for particular species, optimized surveillance should consist of an iterative feedback between local knowledge and model predictions to guide field work and updating the model with new results from field sampling to guide ongoing surveillance and field studies [16,19]. Importantly, to reduce study effort bias going forward, care should be taken to estimate and survey under studied areas and species in addition to newly predicted species, particularly in areas with high rates of human spillover with poorly known hosts. A lack of investment in understanding the basic ecology of *Leishmania* slows research progress by obscuring the major drivers of disease dynamics and spillover transmission to humans. Leishmaniasis surveillance and management would benefit from larger funding mechanisms prioritizing understanding the ecology of this group of pathogens.

In some cases, our host predictions are supported by multiple independent but complementary lines of evidence (summarized below and in S3 Table). Non-specific *Leishmania* antigens, antibodies, or DNA has been detected in the crab-eating racoon (*Procyon cancrivorus*), neotropical river otter (*Lontra longicaudis*), six-banded armadillo (*Euphractus sexcinctus*), coyote *(Canis latrans)*, and the grey short-tailed opossum (*Monodelphis domestica*) [12,68,71,72]. As tests were not parasite species specific, it is unconfirmed if they were infected with zoonotic *Leishmania* species, but our model suggests they may be. Further, *Leishmania*-like flagellates were discovered in Derby's wooly opossum (*Caluromys derbianus*) [71]; however, to our knowledge this species has not been tested for zoonotic *Leishmania* species.

A number of animals predicted to be hosts for *L. (Leishmania)* were also found to co-occur with vectors of *L. (L.) mexicana* in Mexico: the hairy big-eyed bat (*Chiroderma villosum*), Brazilian brown bat (*Eptesicus brasiliensis*), great sac-winged bat (*Saccopteryx bilineata*), tent-making bat (*Uroderma bilobatum*), Mexican woodrat (*Neotoma mexicana*), black footed pygmy rice rat *(Oligoryzomys nigripes)*, Coues's rice rat (*Oryzomys couesi*), fulvous harvest mouse (*Reithrodontomys fulvescens*), nine-banded armadillo (*Dasypus novemcinctus*), lowland paca (*Cuniculus paca*), Virginia opossum (*Didelphis virginiana*), kinkajou (*Potos flavus*), Northern racoon (*Procyon lotor*), white-nose coati (*Nasua narica*), and the margay (*Leopardus wiedii*) [16,73]. Notably, the nine-banded armadillo and margay had the two highest summed Shapley scores as potential *L. (Leishmania)* hosts. The nine-banded armadillo is a known host of *L. (Viannia)* as well as other zoonotic pathogens including the causative agent of leprosy [74].

Host competence (i.e., the ability to become infected and infectious) has been demonstrated in the lab for four of the predicted species: the Virginia opossum (*Didlephis virginiana*), the large vesper mouse (*Calomys callosus*), spix's yellow toothed cavy (*Galea spixii*), and the Mexican free-tailed bat (*Tadarida brasiliensis*) [71,75–77]. Although host competence for these animals has been tested in the laboratory, infection/exposure has not been tested in the wild or animals have not tested positive in the wild (possibly due to minimal sampling of the species). Our model suggests that these animals are also likely to be naturally infected in the wild and thus warrant further investigation. The large vesper mouse is one of the top predicted species for *L. (Viannia)*. Surprisingly, we found no peer-reviewed field studies testing wild large vesper mice for zoonotic *Leishmania* infection. Finally, our model suggests that a handful of species naturally infected with *L. infantum* in Europe are also likely to be exposed to and susceptible to

zoonotic *Leishmania* in the Americas: the red fox (*Vulpes vulpes*), the European hare (*Lepus europaeus*), and the European rabbit (*Oryctolagus cuniculus*) [78–80].

Importantly, while there are 14 species of zoonotic *Leishmania* parasites in the Americas, our analysis divides these species into two subgenera but does not identify *Leishmania* species specific hosts. Future work could leverage the information presented by the current analysis to identify *Leishmania* species specific hosts. Additionally, our analysis identifies animals that are likely to be infected in the wild but does not narrow the host pool down to competent hosts (animals that are likely to be both infected *and* infectious in the wild). Next steps of model development could identify trait profiles of competent hosts that delineate them from non-competent hosts. Notably, [56] used machine learning to predict unrecognized vectors of leishmaniasis. Metabarcoding of sandflies and their blood meals [66,81] may be useful in linking model predictions of vectors and hosts to local transmission cycles.

Overall, models for both subgenera identified a number of new hosts, including but not limited to the nine-banded armadillo, the margay, the large vesper mouse, and the large headed rice rat, that can be targeted for future surveillance and research to determine their ecological significance in leishmaniaisis transmission and spillover to humans. Next steps should include ground truthing the role of newly identified potential hosts in the zoonotic *Leishmania* transmission cycle through field sampling and laboratory experiments, through engagement with local communities, and iteratively updating the current analysis as new information is collected, taking careful consideration to not perpetuate the effects of study bias. Although a cautious first step in host discovery work, our study can immediately inform surveillance efforts and study design. For instance, host predictions may provide a starting point for implication of hosts during local outbreaks of human leishmaniasis by providing a targeted list of likely suspects to sample. While the study of leishmaniasis has historically focused on rodents and small marsupials, our study suggests that capture methods should more commonly be supplemented by bat mist-netting and/or techniques to capture small to medium sized carnivores. Finally, many newly predicted hosts have ranges that extend throughout the southeastern USA. Ecological niche models suggest that climate warming may increase the range of *Leishmania* vectors as far north as Canada and leishmaniasis endemicity within the US was recently confirmed [82–84]. Similarly, the range of predicted hosts for *L. (Viannia)* extends farther south through Argentina than currently known hosts. Our analysis provides tools for monitoring leishmaniasis risk under global change as these hosts could be used as sentinel species for tracking leishmaniasis range expansion. Testing these hosts for leishmaniasis infection beyond the current putative boundaries of zoonotic *Leishmania* ranges would help to evaluate if range expansion has already occurred as well as provide an updated baseline for future monitoring.

Identifying local transmission cycles is key for reducing leishmaniasis spillover and disease burden in people, especially in the face of land-use and climate change, which likely influence the geographic distribution and extent of vector—host—human contact and were strongly associated with novel host predictions in our models. We estimate that there are likely many unknown hosts of leishmaniasis throughout the Americas, highlighting the complexity of the transmission cycle and a need to increase study effort on the ecology and epidemiology of this system.

## Supporting information

**S1 Text. Search methods for recording *Leishmania* host status: We used the search string listed below to run search Web of Science of *Leishmania* hosts.** We ran the initial search on February 12, 2021. We ran additional searches that included the species names of zoonotic

*Leishmania* on February 22, 2021. Our host database was then updated in June 2021 after the publication of (1).
(DOCX)

**S1 Table. Final traits and sources of traits used in the analysis.**
(DOCX)

**S2 Table. Summed Shapley Scores per species for newly predicted species, averages across 100 model iterations (0.05 percentile—0.95 percentile).**
(DOCX)

**S3 Table. Complementary evidence of predicted hosts.**
(DOCX)

**S1 Fig. Importance of each PCoA phylogenetic dimension in predicting *L. (Viannia)* host status.**
(DOCX)

**S2 Fig. Shapley partial dependence plots showing the effect of PCoA dimensions on *L. (Viannia*) host status and distribution of hosts within each order with PCoA dimension values above or below the threshold.**
(DOCX)

**S3 Fig. Features contributions to predictions for the study effort of mammals for animals included in the *L. (Viannia)* analysis.**
(DOCX)

**S4 Fig. Functional response of study effort to mammal traits for animals included in the *L. (Viannia*) analysis.**
(DOCX)

**S5 Fig. Features contributions to predictions for the study effort of mammals for animals included in the *L. (Leishmania)* analysis.**
(DOCX)

**S6 Fig. Functional response of study effort to mammal traits for animals included in the *L. (Leishmania*) analysis.**
(DOCX)

## Acknowledgments

We thank the Mordecai Lab for providing feedback on manuscript drafts.

## Author Contributions

**Conceptualization:** Caroline K. Glidden, Rafaella Albuquerque Silva, Barbara A. Han, Erin A. Mordecai.

**Data curation:** Caroline K. Glidden, Aisling Roya Murran.

**Formal analysis:** Caroline K. Glidden, Aisling Roya Murran, Adrian A. Castellanos, Barbara A. Han.

**Funding acquisition:** Erin A. Mordecai.

**Methodology:** Caroline K. Glidden, Aisling Roya Murran, Rafaella Albuquerque Silva, Adrian A. Castellanos, Barbara A. Han, Erin A. Mordecai.

**Resources:** Erin A. Mordecai.

**Supervision:** Rafaella Albuquerque Silva, Erin A. Mordecai.

**Validation:** Caroline K. Glidden, Rafaella Albuquerque Silva, Adrian A. Castellanos, Barbara A. Han.

**Visualization:** Caroline K. Glidden.

**Writing – original draft:** Caroline K. Glidden, Erin A. Mordecai.

**Writing – review & editing:** Caroline K. Glidden, Aisling Roya Murran, Rafaella Albuquerque Silva, Adrian A. Castellanos, Barbara A. Han, Erin A. Mordecai.

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
