## [Decision Letter · Decision Letter 0]

24 Dec 2022

Dear Dr. Glidden,

Thank you very much for submitting your manuscript "Phylogenetic and biogeographical traits predict unrecognized hosts of zoonotic leishmaniasis" for consideration at PLOS Neglected Tropical Diseases. As with all papers reviewed by the journal, your manuscript was reviewed by members of the editorial board and by several independent reviewers. In light of the reviews (below this email), we would like to invite the resubmission of a significantly-revised version that takes into account the reviewers' comments. 

We cannot make any decision about publication until we have seen the revised manuscript and your response to the reviewers' comments. Your revised manuscript is also likely to be sent to reviewers for further evaluation.

Sincerely,

Alberto Novaes Ramos Jr

Academic Editor

Charles Jaffe

Section Editor

Reviewer's Responses to Questions

**Key Review Criteria Required for Acceptance?**

**Methods**

-Are the objectives of the study clearly articulated with a clear testable hypothesis stated?

-Is the study design appropriate to address the stated objectives?

-Is the population clearly described and appropriate for the hypothesis being tested?

-Is the sample size sufficient to ensure adequate power to address the hypothesis being tested?

-Were correct statistical analysis used to support conclusions?

-Are there concerns about ethical or regulatory requirements being met?

Reviewer #1: It is worth mentioning that I do not work directly with the modeling approach employed by authors and, therefore, I was not able to judge more fully the pertinence of the analyzes carried out. The authors presented the analyzes and explanations for the use and validation of each one of them, which seemed accurate to me. I have two comments to be considered by authors:

Lines 170-173: Authors attributed the same category (0) for animals that were examined and negative for Leishmania infection and for those that was never surveyed. Wouldn't it be better to divide this analysis: the negatives from those that represent a sample void?

Lines 172-173: Authors include in the analysis infection determined by serology. But this technique does not differentiate infections by Leishmania (Leishmania) from Leishmania (Viannia) due to cross-infections. How did the authors deal with this in the separate analysis of the two subgenres?

Reviewer #2: Yes, the methods are adequate.

Reviewer #3: (No Response)

**Results**

-Does the analysis presented match the analysis plan?

-Are the results clearly and completely presented?

-Are the figures (Tables, Images) of sufficient quality for clarity?

Reviewer #1: Figure 3: Put the "Trait Type" caption outside the frame of L. (Viannia) because it serves for the 2 frames. Or put it in both frames

Supplementary Table 2: Add an extra column with the bibliography that defines the gray boxes

Reviewer #2: The results are correct, well ordered and clear. I think there is an excess of supplementary material.

Reviewer #3: (No Response)

**Conclusions**

-Are the conclusions supported by the data presented?

-Are the limitations of analysis clearly described?

-Do the authors discuss how these data can be helpful to advance our understanding of the topic under study?

-Is public health relevance addressed?

Reviewer #1: Authors were very careful and responsible with the conclusions that were presented in the manuscript, clearly presenting and discussing the study limitations.

Reviewer #2: The discussion is very exhaustive, comparing well the data obtained in this study with the literature. In addition, the usefulness of this prediction is well explained.

Reviewer #3: (No Response)

**Editorial and Data Presentation Modifications?**

Reviewer #1: (No Response)

Reviewer #2: The subgenus is called Leishmania (Viannia), not Vianna. Please correct this throughout the text.

Line 84: Please specify order Diptera, family Psychodidae

Line 87: It would be more correct to speak of potential hosts, as the fact that an animal may be susceptible to infection alone cannot be considered a true host. Please, add “potential” before host. 

Line 106: It is not correct to speak of opportunistic sampling in the study of zoonotic Leishmania, as in most cases targeted sampling of the animal populations under study is performed. Please, remove the word “opportunistic”.

Line 335: Please, add the degree symbol (˚)

Line 354: In figure 2, specify L. (Viannia) and L. (Leishmania) and write them in italics in the figure of the subgenera (c). 

Line 376: Please, add the degree symbol (˚)

Lines 409 and 410: Specify the numbers of predicted hosts in each order. 

Line 418: The numbering of the references is missing

Line 440: The numbering of the references is missing

Line 469: Remove “e.g” when talking about references. 

Line 475: Italics and capital letter in “Leishmania”

Lines 499 and 500: The numbering of the references is missing

Lines 510 and 511: The numbering of the references is missing

Line 514: The numbering of the references is missing

Lines 518 and 519: The numbering of the references is missing

Line 544: Put the reference at the end of the sentence, not at the beginning.

Lines 545 and 546: The numbering of the references is missing

The italics are absent in the names of genera and species throughout the reference section.

Supplementary table 3: review the italics and references numbering.

Reviewer #3: (No Response)

**Summary and General Comments**

Reviewer #1: The authors focused on the still poorly knowledge on wild hosts of zoonotic Leishmania species to predict the unknown hosts using machine learning approach. They used trait-profiles of known hosts to identify potentially unknown hosts, especially concerning biogeography, phylogenetic distance, and study effort. Besides the above-mentioned aspects, I have also these comments:

1) Authors should not include their figures in the Introduction. This section should present the state-of-art of the study theme, including previous (and not the current) studies. Figure 1, for example, does not represent human cases, as is cited in line 64. This will lead authors to also revise the figure numbering, because figures should be numbered in the order of appearance in the text. 

2) Introduction (line 88): I suggest considering as "may act as competent hosts". The course of an infection is dependent on factors related to the parasite population, the host, environmental factors, health status and concomitant infections, among others. This means that these 60 species of mammals have already been demonstrated to be able of acting as a reservoir, being source of vector infection, but this characteristic is not permanent, and these animals will not always be able to establish a course of infection that will lead to the transmission of the parasites, i.e., that will lead them to act as a reservoir. Only field studies can define if a putative reservoir is acting as reservoir in a given time-space scale.

3) Oryzomys megacephalus is a homotypic synonym of Hylaeamys megacephalus, which is the correct name.

4) Discussion (line 516): Wouldn't Thrichomys laurentius also be part of this group of competent hosts (doi:10.1371/journal.pntd.0000589) ?

5) There are some references cited in number and other in text. Please revise this throughout the manuscript to fit them in the journal’s format.

Reviewer #2: It is a truly novel and comprehensive study. Perhaps the methods part is too dense, but it is well explained. 

I think it is a good prediction tool, but it would be interesting to compare the prediction with the reality, especially in those areas where, according to the data obtained, the distribution of Leishmania is extended. 

In addition, it can be very useful on a smaller scale, in possible outbreaks of human leishmaniasis that may occur, to look for implicated hosts.

Reviewer #3: General comments:

This is an interesting article that uses machine learning models to identify host taxa involved in Leishmania transmission. This work is important because it provides insight on the potential role of understudied wildlife species in Leishmania transmission. The approaches used are novel and well explained. Below is a list of minor comments/suggestions, mostly focused on needing to strengthen the introduction which reads a bit vague currently, and points of clarification needed in the methods and discussion. One important thing to highlight a bit more is the meaning of these findings for surveillance purposes. Since the predictions are based on infection data with no information on reservoir status for known species for example, cautious recommendations need to be provided. Generally, I found this article compelling and I believe it should be considered for publication once edits have been made and additional information provided. 

Specific comments:

Line 49: might need define "burden" in this context. Suggest providing an estimated number to avoid being vague.

Line 51-52: “In some areas, risk of infection is increasing in geographic extent.” Suggest rewording to be less vague. 

Line 55-57: The summary could be improved to be more specific and focused on the research findings rather than disease background. 

Line 64: It’s unusual to have a figure in the introduction. Suggest moving to the methods or supplementary materials.

Line 65: “Hotspots” is one word.

Lines 69-82: The introduction is quite long. Suggest trimming the introduction starting with this paragraph as the background on the biology of the pathogen is probably not needed in so much detail, but rather highlighting specific background related to the study objectives (spatial-temporal, host phylogenetics, etc) is more important.

Line 70: Spillover is one word.

Line 76: This needs to be referenced properly. More generally, the authors need to check all the references again as there is a lot of switching between number and Chicago style.

Line 86: Can you provide examples of what these interventions would look like?

Line 87: By wildlife hosts, do the authors mean wildlife species? 

Line 92: Same comment as above; it’s not typical to have figures in the introduction

Line 107: What is meant by "broad surveillance"? As in passive surveillance? A reference is also needed here.

Line 110: Suggest changing "dedicate" to "target".

Line 126: Suggest deleting “In brief”.

Line 140-141: This information may be better positioned in the methods section.

Line 148-149: Are these reservoir hosts? I saw the authors explain in the methods but suggest explaining earlier in the introduction.

Line 167: What search string was used and when was it done? Also wondering why the authors use Web of Science here but PubMED for citation numbers? 

Line 172-173: If the authors are using serology data, then they can't technically be talking about infection, only exposure, since with serology data all that is known is that an individual was infected sometime in the past, and no evidence of current infection. Thus, the authors either need to remove the serology data or change the wording from "infected" to "exposed" throughout the article.

Lines 178-180: Would be good to say what percent of animal taxa that included

Line 192: Suggest defining “land-use intensity”.

Lines 211-221: Great explanation. It would be good to have this explanation in the context of other machine learning models (e.g., RF, SVM) for folks less familiar with these models.

Line 222: It would be good to indicate the number of traits that were removed after this step and how the authors decided which ones to include vs. leave out. 

Line 262: Would be good to see the R packages listed somewhere. Additionally, will the authors be sharing their data and code? 

Line 311: I like figure 1, but wondering if the heatmap color needs to be changed as it’s a little challenging to see differences. Maybe try yellow to red or blue-yellow-red?

Line 328: The figures in the supplementary materials showing all 30 features are quite interesting. Might be worth moving them to the main text.

Line 355: Not totally clear why figure 2 is first reported in the introduction.

Line 407-408: Given the large number of new hosts identified, it might be worth reemphasizing that these are just hosts that could be infected (or exposed for the serology) - i.e., not predicted reservoirs. 

Lines 475-477: This will require a date of when the search was done.

Line 486-488: This seems a bit off topic and may derail the readers from the actual goal of the paper. It’s good to raise the issue, but suggest not dwelling on it too much.

Additionally, this paragraph reads like the last or before-last paragraph of the manuscript as it is broadening out, but then the authors dive us back into model result interpretation. So may need to move paragraphs around a bit.

Lines 524-527: This paragraph is so short it’s not really worthy of a paragraph. Suggest expanding, merging with another paragraph, or deleting.

Lines 540-541: So how can these results be used for surveillance? Need to provide clear and cautious guidance with these types of findings. Perhaps make a point of saying that this is "a first step".

PLOS authors have the option to publish the peer review history of their article (what does this mean?). If published, this will include your full peer review and any attached files.

Reviewer #1: No

Reviewer #2: No

Reviewer #3: No
---

## [Decision Letter · Decision Letter 1]

22 Mar 2023

Dear Dr. Glidden,

Thank you very much for submitting your manuscript "Phylogenetic and biogeographical traits predict unrecognized hosts of zoonotic leishmaniasis" for consideration at PLOS Neglected Tropical Diseases. As with all papers reviewed by the journal, your manuscript was reviewed by members of the editorial board and by several independent reviewers. The reviewers appreciated the attention to an important topic. Based on the reviews, we are likely to accept this manuscript for publication, providing that you modify the manuscript according to the review recommendations. 

Sincerely,

Alberto Novaes Ramos Jr

Academic Editor

Charles Jaffe

Section Editor

Reviewer's Responses to Questions

**Key Review Criteria Required for Acceptance?**

**Methods**

-Are the objectives of the study clearly articulated with a clear testable hypothesis stated?

-Is the study design appropriate to address the stated objectives?

-Is the population clearly described and appropriate for the hypothesis being tested?

-Is the sample size sufficient to ensure adequate power to address the hypothesis being tested?

-Were correct statistical analysis used to support conclusions?

-Are there concerns about ethical or regulatory requirements being met?

Reviewer #1: (No Response)

Reviewer #3: (No Response)

**Results**

-Does the analysis presented match the analysis plan?

-Are the results clearly and completely presented?

-Are the figures (Tables, Images) of sufficient quality for clarity?

Reviewer #1: (No Response)

Reviewer #3: (No Response)

**Conclusions**

-Are the conclusions supported by the data presented?

-Are the limitations of analysis clearly described?

-Do the authors discuss how these data can be helpful to advance our understanding of the topic under study?

-Is public health relevance addressed?

Reviewer #1: (No Response)

Reviewer #3: (No Response)

**Editorial and Data Presentation Modifications?**

Reviewer #1: (No Response)

Reviewer #3: (No Response)

**Summary and General Comments**

Reviewer #1: Authors satisfactorily answered the reviewers’ comments and modified the manuscript as requested. However, I suggest authors to include the answer in which they justify the use of “non-positive” instead of “true-negative” (first comment of Reviewer 1) in the methos or even discussion of the manuscript. 

Other minor corrections are:

1) Line 220: Exclude “from”

2) Figure 1 is divided into a,b,c,d. But in the legend and in the text e, f, g, h are still mentioned. Review this.

3) Line 409: Revise “there is are is high”

4) Line 552: I suggest changing “Bolsonaro” for "federal administration from 2019-2022"

5) Line 590: “Didelphis”, not “Didlephis”

Reviewer #3: The authors have addressed the issues raised in the previous review.

PLOS authors have the option to publish the peer review history of their article (what does this mean?). If published, this will include your full peer review and any attached files.

Reviewer #1: No

Reviewer #3: No

Figure Files:

Data Requirements:

Reproducibility:

References

---

## [Decision Letter · Decision Letter 2]

1 May 2023

Dear Dr. Glidden,

We are pleased to inform you that your manuscript 'Phylogenetic and biogeographical traits predict unrecognized hosts of zoonotic leishmaniasis' has been provisionally accepted for publication in PLOS Neglected Tropical Diseases.

Best regards,

Alberto Novaes Ramos Jr

Academic Editor

Charles Jaffe

Section Editor

Reviewer's Responses to Questions

**Key Review Criteria Required for Acceptance?**

**Methods**

-Are the objectives of the study clearly articulated with a clear testable hypothesis stated?

-Is the study design appropriate to address the stated objectives?

-Is the population clearly described and appropriate for the hypothesis being tested?

-Is the sample size sufficient to ensure adequate power to address the hypothesis being tested?

-Were correct statistical analysis used to support conclusions?

-Are there concerns about ethical or regulatory requirements being met?

Reviewer #1: (No Response)

Reviewer #3: (No Response)

**Results**

-Does the analysis presented match the analysis plan?

-Are the results clearly and completely presented?

-Are the figures (Tables, Images) of sufficient quality for clarity?

Reviewer #1: (No Response)

Reviewer #3: (No Response)

**Conclusions**

-Are the conclusions supported by the data presented?

-Are the limitations of analysis clearly described?

-Do the authors discuss how these data can be helpful to advance our understanding of the topic under study?

-Is public health relevance addressed?

Reviewer #1: (No Response)

Reviewer #3: (No Response)

**Editorial and Data Presentation Modifications?**

Reviewer #1: (No Response)

Reviewer #3: (No Response)

**Summary and General Comments**

Reviewer #1: A final typographic correction has to be made before accptance: 2022 instead of 202 in line 590

Reviewer #3: (No Response)

PLOS authors have the option to publish the peer review history of their article (what does this mean?). If published, this will include your full peer review and any attached files.

Reviewer #1: No

Reviewer #3: No

---

## [Editor Report · Acceptance letter]

15 May 2023

Dear Dr. Glidden,

We are delighted to inform you that your manuscript, "Phylogenetic and biogeographical traits predict unrecognized hosts of zoonotic leishmaniasis," has been formally accepted for publication in PLOS Neglected Tropical Diseases.

Best regards,

Shaden Kamhawi

co-Editor-in-Chief

Paul Brindley

co-Editor-in-Chief
